# Chrono-Nutritional Patterns, Medical Comorbidities, and Psychological Status in Patients with Severe Obesity

**DOI:** 10.3390/nu15235003

**Published:** 2023-12-03

**Authors:** Silvia Bettini, Sami Schiff, Enrico Carraro, Chiara Callegari, Beatrice Gusella, Giulia Maria Pontesilli, Matteo D’Angelo, Valeria Baldan, Alessandra Zattarin, Giulia Romanelli, Paolo Angeli, Paolo Girardi, Paolo Spinella, Roberto Vettor, Luca Busetto

**Affiliations:** 1Department of Medicine, University of Padova, 35128 Padova, Italy; sami.schiff@unipd.it (S.S.); chiara.callegari.05@gmail.com (C.C.); beatrice.gusella@gmail.com (B.G.); pontesilligm@gmail.com (G.M.P.); matteo.dangelo@aopd.veneto.it (M.D.); valeria.baldan@aopd.veneto.it (V.B.); alessandra.zattarin@aopd.veneto.it (A.Z.); giulia.romanelli@aopd.veneto.it (G.R.); pangeli@unipd.it (P.A.); paolo.spinella@unipd.it (P.S.); roberto.vettor@unipd.it (R.V.); luca.busetto@unipd.it (L.B.); 2Internal Medicine Unit 3, Padova University Hospital, 35128 Padova, Italy; 3Internal Medicine Unit 5, Padova University Hospital, 35128 Padova, Italy; 4Department of Statistical Sciences, University of Padova, 35121 Padova, Italy; enrico.carraro.10@phd.unipd.it; 5Department of General Medicine, Santa Maria della Misericordia Hospital, 45100 Rovigo, Italy; 6Dietetics and Clinical Nutrition Unit, Padova University Hospital, 35128 Padova, Italy; 7Department of Environmental Sciences, Informatics and Statistics, Ca’ Foscari University of Venice, 30172 Venezia, Italy; paolo.girardi@unive.it

**Keywords:** obesity, chrono-nutrition, dietary patterns, psychological parameters, dietary habits, physical and mental health

## Abstract

Chrono-nutrition studies dietary habits and their role in the onset of metabolic diseases. The aim of this study is to describe chrono-nutritional patterns based on the analysis of the eating habits of patients with severe obesity during the 24-h cycle and investigate a possible relationship between these profiles, the comorbidities, and the psychological status. From the overall evaluation of the chrono-nutritional profiles of 173 patients with severe obesity, four predominant eating patterns were obtained with a refined statistical model. A regression analysis was performed to determine the relationship between chrono-nutritional patterns, medical comorbidities, and psychological status. Profile 1 was the most frequent (46.2%) and characterised by the regular presence of the three main meals. The distribution of the chrono-nutritional profiles did not vary with BMI. Chrono-nutritional profiles affected predominantly psychological variables, with lower performances among chrono-nutritional profiles 3 (to eat during all the 24-h, with nibbling and snacking also during the night) and 4 (like the fourth but without night-eating). This finding could be useful in the assessment and treatment of patients with obesity, allowing the identification of patients with a higher probability of suffering from a psychopathological condition simply by knowing the patients’ dietary profiles.

## 1. Introduction

Obesity is a chronic relapsing disease associated with a wide range of other non-communicable diseases, including metabolic, functional, and psychological disorders [1,2]. Circadian rhythms are essential regulators of the physiology of several biological processes. A great amount of evidence suggests that meal timing can affect physiological processes, including the sleep-wake cycle, body temperature, metabolism, blood pressure, and hormonal levels, and can impact health, being a major contributor to obesity and related comorbidities [3,4,5,6]. In this context, chrono-nutrition is an emerging field of research involved in the study of food intake in coordination with the body’s daily rhythms and their role in the onset of metabolic diseases, particularly obesity [7,8]. In chrono-nutritional studies, it was demonstrated that more than 50% of the population (non-shiftwork) eats for approximately 15 h a day and that increasing time to eat with shorter overnight fasts contributes to increased energy intake [9,10].

However, analysis of the circadian pattern of people living with obesity demonstrated much more complex and erratic nutritional habits. Gill et al. [11] developed a smartphone app to monitor food intake in healthy adults, revealing frequent and erratic daily eating patterns rather than the self-reported three daily meals, leading to metabolic problems.

The aim of this study is therefore to describe chrono-nutritional patterns based on the analysis of the eating habits of patients with obesity during the 24 h cycle and investigate a possible relation between these profiles and the medical comorbidities and psychological status. This finding could be useful in the assessment and treatment of patients with obesity, allowing the identification of patients with a higher probability of comorbidities or suffering from psychopathological symptomatology, which may contribute to dysfunctional, unhealthy eating, simply by knowing the patients’ chrono-nutritional profile.

## 2. Materials and Methods

### 2.1. Patients’ Multi-Disciplinary Assessment

#### 2.1.1. Patients: Inclusion/Exclusion Criteria

In this observational study, 193 patients with severe obesity were enrolled in the Centre for the Study and the Integrated Treatment of Obesity at the University Hospital of Padua, Italy. These patients qualified for bariatric surgery according to the European guidelines for obesity management in adults [12]. In preparation for surgery, an extensive multi-disciplinary assessment was performed, including medical assessment, biochemical testing, psychological evaluation with psychometric tests, and dietetic assessment. All patients received nutritional education and counselling. A single database collected all the information obtained during the multi-disciplinary assessment. Only patients with complete information on all the domains of the multidisciplinary evaluation (medical and dietetic assessment, biochemical testing, and psychological evaluation with psychometric tests) were finally included in this study; thus, we recruited 173 patients. All patients gave their informed consent in accordance with the Declaration of Helsinki. The protocol was approved by the Ethics Committee of Padua for Clinical Research (2892P, 10 June 2013).

#### 2.1.2. Clinical Evaluation

Clinical data includes anthropometric measurements, sex, age, education, exercise, alcohol and smoking habits, the presence of hypertension, diabetes, and dyslipidaemia. Blood testing was performed to assess the presence of undiagnosed conditions as well as OSA testing, if deemed necessary. In particular, participants also responded to a series of questions assessing alcohol consumption (reporting the number of daily alcohol units and weekly frequency), smoking habits (number of cigarettes per day), and physical activity (type of activity, number of times per week and duration of each session in minutes).

#### 2.1.3. Anthropometric Measurements

All anthropometric measurements were taken with participants wearing only light clothes without shoes. Height was measured to the nearest 0.01 m using a stadiometer. Body weight was determined to the nearest 0.1 kg using a calibrated balance beam scale. BMI was calculated as weight (kg) divided by height squared (m^2^).

#### 2.1.4. Biochemical Assessment

Patients’ blood were tested to evaluate the following: fasting plasma glucose (FPG), insulin, lipid profile [total cholesterol (TC), High-Density Lipoprotein-cholesterol (HDL), Low-Density Lipoprotein-cholesterol (LDL), and triglycerides (TG)]. All patients were in a fasted state (8-h fast) when being tested. Blood test samples were stored at −20 °C until analysis. All biochemical blood analyses were performed with standard diagnostic kits according to the WHO First International Reference Standard. LDL cholesterol was calculated according to Friedewald [13]. Patients without a diabetes diagnosis underwent a 3-h oral glucose tolerance test (OGTT), which monitored blood glucose and insulin plasma levels after a 75 g glucose load. Insulin resistance was evaluated by calculating the homeostasis model assessment of insulin-resistance index (HOMA): [fasting serum insulin (μU/mL) × FPG (mmol/L)]/22.5. However, HOMA was not calculated for patients who were being treated with insulin at that point in time.

#### 2.1.5. Nutritional Assessment 

During the dietetic assessment and counselling session, the usual dietary pattern of the patients was assessed through a 24-h dietary recall referring to the working day preceding the visit. The analysis of the recall provided information on times of eating and dietary habits which were used to create dietary patterns in statistical analysis. The 11 meal types that were identified in the 24-h dietary recall included breakfast, morning snack, morning nibbling, lunch, afternoon snack, afternoon nibbling, dinner, post-dinner snack, post-dinner nibbling, late-night snack, and late-night nibbling. Breakfast, lunch, and dinner represented the main meals. It is crucial to differentiate snacking from nibbling because both of them were consumed between meals: snacking refers to a light and quick, smaller meal placed in between main meals, whereas nibbling, as described in the 17th version of the Eating Disorder Examination, refers to unplanned and constant eating between meals [14].

#### 2.1.6. Psychometric Parameters

Psychometric data were collected via the following psychological tests: Short Form Health Survey (SF-36), Symptom Checklist-90-Revised (SCL-90-R), Eating Attitude Test 26 (EAT-26), Binge Eating Scale (BES), Yale Food Addiction Scale (Y-FAS), and Barratt Impulsiveness Scale-11 (BIS-11).

SF-36 evaluates the individual’s perception of their physical and mental health as well as their quality of life [15,16]. SCL-90-R assesses the presence and the intensity of psychopathological externalised and internalised symptoms [17,18]. This questionnaire allows us to explore a variety of symptoms and includes different subscales: somatisation, interpersonal sensitivity, obsessive-compulsive, depressive, anxiety, and phobic symptoms, hostility, psychoticism, and paranoid ideation. The SCL-90-R includes items assessing appetite and sleep disorders [17,18]. EAT-26 is a 26-item questionnaire that aids in eating disorder screening [19], while BES is composed of 16 items assessing behavioural characteristics, and emotional and cognitive responses related to binge eating [20,21]. Y-FAS is a 25-point questionnaire based on DSM-IV [22] codes for substance dependence criteria to assess food addiction [23,24,25] and BIS-11, which is a 30-item questionnaire designed to assess the personality/behavioural construct of impulsiveness [26,27]. In this work for each test, the pertinent scoring was computed and the following scales were used in data analysis: SCL-90-R Global Symptoms Intensity (SCL-90-R_GSI); SF-36 Physical and Mental Health (PH and MH); EAT-26; BES; BIS-11 total scores; and Y-FAS symptoms number.

### 2.2. Statistical Analysis

#### 2.2.1. Descriptive Analysis

The variables of interest were summarised by frequency tables and figures (frequency for categorical variables, median, and InterQuartile Range (IQR) for continuous variables). Non-parametric tests (Wilcoxon or Kruskal-Wallis tests) were computed to compare the distribution of non-gaussian quantitative variables across strata. Categorical variables were compared using the chi-squared or Fisher’s exact test, where the expected frequencies in any combination were less than 10. Statistical significance was assumed at the 5% level. All descriptive analysis, clustering procedures, and regression models were performed using R Statistical Software, v.4.3 [28] using ad-hoc functions and several packages.

#### 2.2.2. Data Imputation

In order to complete the missing values for some subjects’ records, the Multivariate Imputation by Chained Equations (MICE) algorithm [29] was used under the Missing at Random (MAR) assumption. Among the 173 patients collected, the variables with the relative percentage of missing data were: dyslipidemia 14.5%, shift work 0.6%, BIS total 2.9%, total cholesterol 14.5%, HDL 15.0%, LDL 15.0%, triglycerides 14.5%, and HOMA index 15.0%. The imputation process was performed using predictive mean matching since the variables had a continuous distribution, except for dyslipidemia and the presence of shift work, where a binary logistic regression was employed. The imputation algorithm was iterated 50 times, and the convergence was graphically evaluated by inspecting the variability of the imputed values.

#### 2.2.3. Chrono Nutrition Data—A Functional Data Analysis

The 24-h dietary recall was analysed with the aim to produce a clustering of patients having a similar pattern of daily nutrition in terms of the time in which they consumed foods. In particular, as stated above, information was collected about the habit of having 11 different types of meals or food contacts: breakfast (7:00 to 8:00), morning nibbling (9:00 to 12:00), morning snack (10:00 to 11:00), lunch (12:00 to 13:00), afternoon nibbling (14:00 to 18:00), afternoon snack (16:00 to 17:00), dinner (19:00 to 20:00), evening nibbling (21:00 to 24:00), evening snack (21:00 to 22:00), night nibbling (1:00 to 6:00), and night snack (3:00 to 4:00). For each individual, a chrono-nutritional profile was therefore built, denoted by the presence or absence of these pre-specified food contacts. Considering this particular set of data, standard clustering techniques based on dissimilarity matrices fail to perform a correct classification because temporal features are not commonly taken into account in the clustering algorithm. A more informative strategy started by matching each meal with the relative time of consumption: for each subject, a step function was obtained with a value equal to 1 in the case of meal and 0 otherwise. In order to classify similar step functions, a strategy based on the clustering of functional data was adopted. The probability of having a meal was denoted by the step-function *f*(*t*) during a certain *time t* = *1, ..., 24*, and our strategy was to approximate this function by means of a Fourier basis of order 2 as follows
(1)$$f(t)=\alpha +\sum_{n=1}^{2}\left(a_{n}coscos\;|2\pi nt/24|+b_{n}sinsin\;|2\pi nt/24|\right)$$where α is the intercept and $a_{n}$ and $b_{n}$ are the coefficients of the Fourier basis. Each patient reported the value *y*(*t*), denoting the absence or presence of a meal at the time *t*. We can
*y*(*t*) = *f*(*t*) + *ϵ*(*t*)(2)where $\epsilon \left(t\right)$ is an error term with a mean of 0 and a constant variance.

The function *f*(*t*) can be usually estimated by an Ordinary Least Squares (OLS) procedure. In this implementation, a penalisation term was introduced in the loss function as follows
(3)$$\sum_{t=1}^{24}{\left(f(t)-y(t)\right)}^{2}+\lambda \int_{1}^{24}[{D^{2}f(t)]}^{2}dt\;$$where λ is a roughness penalty applied to the total curvature measure $\int_{1}^{24}[{D^{2}f(t)]}^{2}dt$.

The value of λ was chosen by means of an overall generalised cross-validation strategy [30]. The analysis of functional data was performed by the R package *fda*. However, it is important to report that this procedure does not ensure *f*(*t*) stays between 0 and 1, but from a practical point of view, it obtains reasonable results. Other strategies require the use of penalised logistic regression and logit transformation, but the resulting curves can suffer from flatness due to the presence of an over-smoothing effect and a potential bias due to the small sample size.

Our procedure estimated the smoothed probability of having a meal during the day was obtained for each subject ($\hat{f_{i}}\left(t\right),\;i=1,\ldots ,\;173)$. An example of a chrono-nutritional time series for a patient and the relative estimated curve was reported in Appendix A. In order to cluster together subjects with a similar chrono-nutritional profile, we performed clustering by means of a functional High-Dimensional Data Clustering (*funHDDC*) algorithm, which allows the clustering of functional univariate and multivariate data [31]. The number of clusters was selected by minimizing the Bayesian Information Criterion (BIC) value. The classification obtained in this section was then used in the next regression analysis.

#### 2.2.4. Quantile and Logistic Regression

Since most of the marginal distribution of physiological values and psychological scores were asymmetrically distributed and reported the presence of outliers, we evaluated which factors influenced the set of outcomes of interest by means of a quantile regression model performed on the quantile 0.5, which is the median value. Variables were included on the basis of the AIC index on a backward approach starting from a full model, which included gender (male, female), age in continuous form, educational level (middle school, high school, degree or higher), physical activity (no, yes), alcohol consumption (no, yes), smoking habit (yes, no), body mass index (BMI, three categories based on tertiles), and chrono-nutritional profile. Only for psychological scores, in the starting model the variables related to hypertension (no, yes), diabetes (no, pre-diabetes, yes), and dyslipidemia (no, yes) were included. The model results were presented as coefficient estimates and their relative 95% Coefficient Intervals (95% CI). The estimation of the coefficients was performed using the R package *quantreg* [32].

Clinical variables such as dyslipidemia, hypertension, and diabetes were distributed on a binary scale (absence/presence) justifying the use of a logistic binary regression model. The aim of this regression model was to verify which factors influenced the presence of these outcomes. Variables were included based on the AIC index in a backward selection procedure starting from a full model, including the same variables used in the model based on the quantile regression. Results were presented by means of Odds Ratios (OR) by means of an exponential transformation of the model coefficients with their relative 95% Coefficients Intervals (95% CI).

## 3. Results

### 3.1. Chrono-Nutritional Profiling

The generalised cross-validation (GCV) was applied to estimate the value of λ in order to reduce the roughness of the functional profiles for each individual (an example of application was reported in Appendix A) and the estimated chrono-nutritional curves at the individual levels were reported in Figure 1a. The funHDDC algorithm applied to the individual curves suggested the presence of four clusters according to the inimization of the BIC value. The estimated curves belonging to each discovered chrono-nutritional profile are reported in Figure 1b. 

The first most frequent profile (Profile 1, N = 80, 46.2%) was characterised by the regular presence of the three main meals (breakfast, lunch, and dinner), by the absence of nibbling, and by a moderate propensity to snack during the morning and the afternoon, but not after dinner or at night. The second profile (Profile 2, N = 11, 6.4%) was formed by patients who regularly had all three main meals and snacks (in the morning, in the afternoon, and in the evening), but not nibbling during the day or any food contacts during the night. Conversely, the third profile (Profile 3, N = 55, 31.8%) was characterised by a constant propensity to eat during all the 24-h, with nibbling and snacking during the morning (nibbling 15%, snacking 47%), the afternoon (nibbling 40%, snacking 68%), the evening (nibbling 13%, snacking 46%), and even the night (nibbling 11%, snacking 13%). Finally, the fourth profile (Profile 4, N = 27, 15.6%) was similar to the previous one, but with a higher propensity to eat out of the meals during the day and after dinner, but not at night. The difference between the percentages of food contacts among the estimated profiles was highly statistically significant except for breakfast, lunch, and dinner (Table 1).

### 3.2. Descriptive Analysis

The median age of the patients was 46 years (IQR:39-55) ranging from 17 to 71 years old. The BMI ranged from 32.9 kg/m^2^ to 80.3 kg/m^2^, with an average value of 44 kg/m^2^ (median 43 kg/m^2^). The sample reported a marked predominance of females (68%) and the educational level was mainly between the middle and high school (42 and 43%, respectively). Only 17% of respondents reported doing physical activity as well as a limited percentage of those reporting alcohol consumption (14%), or a smoking habit (19%). About 18% of patients worked in shifts. BMI levels were negatively correlated with gender (*p* = 0.011, Table 2), since in the last tertile of BMI (46.2–80.3 kg/m^2^), the percentage of females decreased to 53%. The other main characteristics of the sample were not influenced by the BMI category (Table 2). In particular, the distribution of the chrono-nutritional profiles does not vary according to BMI levels.

The clinical status and the results of psychological testing overall and by BMI categories were reported in Table 3. Quite half of the sample had a diagnosis of hypertension, while diabetes was reported in 18%. Pre-diabetes occurred in 40% of the patients. Most patients suffered from dyslipidemia (60%), but only 10% used statins at the time of enrollment. The SF-36 index reported an overall median value of 52 (IQR: 35–71) and 55 (IQR: 39–72) for the physical and mental health components, respectively. The Y-FAS score reported a central point of 2 (IQR: 1–4). The EAT-26 score had an overall value of 8 points (IQR: 3–14), while the total BIS score reported a median value of 59 points (IQR: 53–67). The BES score reported a median equal to 12 (IQR: 6–19). The frequency of dyslipidemia, as well as total cholesterol, HDL, and HOMA levels, were negatively influenced by BMI values. For psychological scores, only the physical component of the SF-36 and the EAT-26 score demonstrated a significant association with BMI values (Table 3). 

The distribution of the main patient’s characteristics and the considered outcomes across chrono-nutritional profiles were reported in Appendix A, respectively. Educational level, physical activity, alcohol consumption, shift work, and belonging to a particular BMI category did not differ among the 4 profiles; only smoking habit was less frequent among profiles 3 and 4 (*p* = 0.038) (Appendix A). While any difference was described based on the presence of comorbidities, patients belonged to chrono nutritional profiles 3 and 4 presented the worst performance in psychological scores, in particular, SCL90-R_GSI (*p* = 0.001), Y-FAS score (*p* = 0.010), BIS-11 score (0.034), and BES (*p* = 0.002) (Appendix A).

### 3.3. Quantile and Logistic Regression Models

ORs for the presence of comorbidities (hypertension, dyslipidemia, and diabetes) were estimated by three different logistic regression models and reported in Table 4. Male patients reported an increased risk of hypertension (OR: 2.21, 95%CI: 1.08–4.60), dyslipidemia (OR: 11.7, 95%CI: 4.90–33.0), and diabetes (OR: 2.99, 95%CI: 1.26–7.24) as compared to female patients. Age influenced the frequency of hypertension (OR: 1.09, 95%CI: 1.06–1.14) and diabetes (OR: 1.05, 95%CI: 1.00–1.09). The consumption of alcohol was associated with a lower risk of hypertension (OR: 0.34, 95%CI: 0.12–0.88). Doing physical activity decreased the risk of a diagnosis of diabetes (OR:0.31, 95%CI: 0.05–1.20), while the opposite effect was reported by the BMI, with increasing risk with an increase in the BMI tertiles (2nd tertile OR: 1.61, 3rd tertile OR: 3.12, 95%CI: 1.07–10.1). No associations were found between chrono-nutritional profiles and the risk of comorbidities in the logistic regression model (Table 4). 

The estimated coefficients related to the quantile regression on some metabolic parameters are reported in Table 5. HDL increased in the presence of smoking habits (0.14, 95%CI: 0.02–0.27), while decreasing in the second (−0.16, 95%CI:−0.32–−0.00), and third tertiles (−0.16, 95%CI: −0.31–−0.00) of the BMI category with respect to the first tertile.

TG concentration increased with age (+0.01 mmol/L in the median for a 1-year increase). Alcohol consumption and the BMI category increased the TG concentration. We observed a TG increase among subjects in the chrono-nutritional Profile 3 with respect to those in the Profile 1 (+0.19, 95%CI: −0.12–0.49). The HOMA index was influenced by gender (+2.62 points among males vs. females, 95%CI: 0.97–4.26) and the last BMI category (+1.90 in the 3rd tertile vs. the 1st tertile of BMI, 95%CI: 0.45–3.47). No associations were found between chrono-nutritional profiles, HDL concentration, and HOMA index (Table 5). 

Table 6 reports the coefficients estimated by quantile regression models for psychological scores. As expected, the physical component of quality of life (SF-36) was negatively associated with age and a low level of education, while a relative intermediate BMI level [41.2–46.2] Kg/m^2^ reported a positive effect with respect to those in the first tertile [<41.2] Kg/m^2^. In addition, in both physical and mental health status, a positive effect was exerted by alcohol consumption, while a negative variation in such indices was shown by patients with dyslipidemia.

Doing physical activity was associated with better mental and physical quality of life in terms of SF-36, but with lower impulsiveness (BIS-11 scale) and symptoms of eating disorder (EAT-26). The smoking habit was significantly associated with a decrease in SF-36 Mental Health. Alcohol consumption was associated with higher scores on the food addiction scale (Y-FAS). As observed in the SF-36 scales, paradoxically, higher BMI tertiles were risk factors for lower EAT26, BIS-11, BES, and SCL90-R_GSI scores. As far as the chrono-nutritional profiles (Table 6), belonging to Profile 3, characterised by a constant propensity to eat during all the 24-h, including the night, are associated with significant variations in many psychological scores. In particular, patients in Profile 3, as compared to patients in Profile 1 (regular presence of the three main meals, absence of nibbling, and presence of night eating), were characterised by a lower perceived physical and mental quality of life (SF-36 MH and FH), more intense psychopathological symptoms (SCL-90-R_GSI), more symptoms and concerns characteristic of eating disorders (EAT-26), increased impulsiveness (BIS-11), binge eating (BES), and food addiction (Y-FAS). Patients belonging to Profile 4, similar to Profile 3, but without night eating, were characterised by lower perceived physical health, more symptoms and concerns characteristic of eating disorders (EAT-26), and more binge eating (BES) as compared to patients in Profile 1. 

## 4. Discussion

The emerging field of chrono-nutrition focuses on the interaction between circadian rhythm, nutrition, and metabolism. This discipline can provide further insight into the factors associated with obesity, as one of its main goals is to explore the association between nutritional profile, BMI, and weight-related comorbidities [7,8]. Patients’ dietary habits seen through a chrono-nutrition lens can shed light on the development of these obesity-associated comorbidities, and possibly, their management [7,8,9,10].

The primary aim of this study was to identify the main temporal eating patterns as a function of the probability of having food contact during different moments in the 24-h of 173 patients with severe obesity, and secondary, to explore the association of these patterns with clinical comorbidities and psychological variables. Using this approach, four main temporal eating patterns were identified. Subjects in Profile 1 reported having mostly three daily meals (“meal eaters”). Patients in Profile 2 had three main meals and a couple of snacks in between (“meal and snack eaters)”. Profile 3 consisted of patients who reported having many different kinds of meals, including nibbling, predominantly in the daytime but with a certain percentage of eating during the night (“continuous day and night eaters”). Profile 4 had the same pattern as the previous one, but without nibbling occurring during the night (“continuous day eaters”). It is important to note that these last two groups consisted of nearly half (47%) of all the individuals included in the study: their diet was unstructured, chaotic, and in sharp contrast with the recommended eating patterns followed by Profiles 1 and 2. 

The most interesting and relevant result of our study is that the eating patterns described in our sample correlate strongly with the psychological profile of the patients. In particular, “continuous day and night eaters” (Profile 3), as compared to “meal eaters” (Profile 1), were characterised by a lower perceived physical and mental quality of life (SF-36), more intense psychopathological symptomatology (SCL-90-R_GSI), more symptoms and concerns characteristics of eating disorders (EAT-26), increased impulsiveness (BIS-11), higher binge eating symptoms (BES), and food addiction scores (Y-FAS). On the other hand, “continuous day eaters” (Profile 4) were characterised by a lower perceived physical but not the mental quality of life (SF-36), higher symptoms and concerns characteristics of eating disorders (EAT-26), and more binge eating symptoms (BES), as compared to “meal eaters” (Profile 1), but they were not characterised by increased impulsiveness and food addiction (Y-FAS). Overall, these results suggest that “continuous day and night eaters” show symptoms of disordered eating like “continuous day eaters” but with worse mental health status and perception and higher impulsivity. This confirms previous findings [33], where patients with a diagnosis of both binge eating and night eating syndrome suffered from more severe psychopathology compared to those with a single diagnosis. This seems to be more related to nocturnal behaviour itself than to an increased BMI due to overeating. In fact, no significant difference in BMI between Profile 3 and Profile 4 was found.

Moreover, patients characterised by more disorganised eating patterns (Profile 3 and 4) did not present differences in obesity severity (BMI categories) compared to Profile 1. These findings could be explained by the methods used to define the chrono-nutritional patterns, where the Authors’ aim was the identification of temporal eating patterns, independent of the caloric intake, the types of foods consumed, portion sizes, and macro-nutrient distribution. Conversely, smoking was less frequent in Profiles 3 and 4. A possible explanation for this was the frequent daytime and night-time nibbling behaviour of subjects in these clusters, who may be using food as a coping mechanism for stress instead of other behaviours such as smoking. 

Considering obesity-related diseases (hypertension, diabetes, dyslipidaemia), their prevalence was much more associated with age, gender, and BMI, as expected. However, no associations were found between chrono-nutritional profiles and comorbidities or HDL concentration and HOMA index. It was only observed a TG increase among subjects in Profile 3 with respect to those in Profile 1. This is in contrast with previous observations linking chronotype with obesity severity and complications [3,4,5,6]. Chronotype is the attitude of a subject in determining individual circadian preference in behavioural and biological rhythm relative to the external light-dark cycle. It was previously demonstrated that individuals with evening chronotype are more prone to follow unhealthy lifestyles and they have higher BMI values in a sample of middle-aged people enrolled in a campaign to prevent obesity [34]. Furthermore, evening chronotype was associated with a significantly higher risk of type 2 diabetes in postmenopausal women with obesity [35]. The lack of correlation between Profile 3 and 4 and the HOMA index or the presence of diabetes may be due to several reasons, including a small sample size, heterogeneity of the cohort, and, as we stated above, the macro-nutrient distribution was not measured. 

At the clinical level, our observations suggest that the simple recall of the eating pattern could permit the identification of a relevant subgroup of patients in which eating is mostly associated with mental status and not driven simplistically by the homeostatic cycle of hunger and satiation. The identification of this subgroup of patients could be relevant to the choice of treatment modalities and their outcomes. Other studies revealed that the change of the eating pattern, by reducing eating duration in the 24 h reduced body weight and improved sleep [11]. Moreover, dysfunctional eating has been associated, albeit not consistently, with weight loss after bariatric surgery [36,37,38,39,40,41,42]. Emotional eating has been shown to negatively predict weight loss in patients with morbid obesity following bariatric surgery [36,37,38] or weight loss programs [38]. On the other hand, if patients with psychiatric disorders (particularly depression and/or personality disorders) preceding surgery showed a less successful outcome an eating disorder was not a negative predictor of weight loss [40]. Regarding food addiction, one study showed that the presence of food addiction negatively predicts weight loss 12-month post-surgery [39], but the other two studies did not confirm this association [36,37,38,39,40,41,42]. Impulsivity, among other psychological factors, might also negatively affect post-surgery weight loss [43]. Notwithstanding these controversial results, the pre-operative identification of patients having a high risk of poor weight loss after surgery could help to focus psychological interventions on patients deserving more psychological support.

Our study presents different limitations. Overall, the study suffered from a small sample size: despite the presence of 173 patients, regression analysis on dichotomous variables can be difficult because of the case sparsity. In fact, this latter effect particularly affected the regression analysis with clinical outcomes. The sample consists of patients with severe obesity candidates for bariatric surgery: while this provides valuable insights into this specific population, it might not be directly applicable to individuals with different obesity levels or the general population. For the same reason, comorbidity prevalence is higher in our sample (44% hypertensive subjects, 40% pre-diabetic, and 18% diabetic subjects), and definitely higher than in the general Italian population [44]. Educational level was also lower in the sample than in the general Italian population, with only 15% of subjects having a university degree, which is supported by epidemiological findings correlating obesity and lack of education. Moreover, we use the 24-h recall as a dietary assessment method and this might not represent the real picture of the dietary patterns. In fact, we describe nutritional patterns based on the probability of having a food contact during different moments in the 24-h and differentiating between main meals, snacking, and nibbling. We were not able to collect more detailed descriptions of these patterns, including the types of foods consumed, portion sizes, calorie counting, and macro-nutrients distribution, which could have enhanced the description of the different nutritional profiles. Finally, in this study, data on sleeping duration were not collected but patients in Profile 3 (“continuous day and night eaters”), representing the more affected psychological profile, evidently consisted of patients with shorter sleep duration.

## 5. Conclusions

In conclusion, this study drew attention to the importance of chrono-nutritional analysis in patients with obesity, when evaluating their overall health status. The analysis of the temporal eating pattern can help to identify a group of patients at higher risk of developing or suffering from psychological and behavioural issues. In particular, patients with disorganised, confused, and unbalanced eating patterns and habits are characterised by a low perceived quality of life, higher mental distress, a tendency to binge eat and presenting symptoms of food addiction, and higher impulsiveness. We confirmed that the psychological issues in the development of obesity [45] are of crucial importance due to their relationship with nutritional habits, especially for the evaluation of obesity at the individual level. Nevertheless, it is important to note that the study establishes associations between chrono-nutritional patterns and psychological status, which does not necessarily imply causation. Further explanation is needed to determine causative relationships. Finally, this study highlights the importance of a complete evaluation of patients’ nutritional profile in all patients with obesity, including the candidates for bariatric surgery, as a helpful risk predictor of psychophysical health and weight loss achievement.

While the relationship between time, food intake, and nutrition quality needs further investigation in future studies, our results highlight how identifying chrono-nutritional temporal patterns according to this new stratification could be a diagnostic approach for improving the therapeutic choice to obtain weight loss. 

## Figures and Tables

**Figure 1 nutrients-15-05003-f001:**
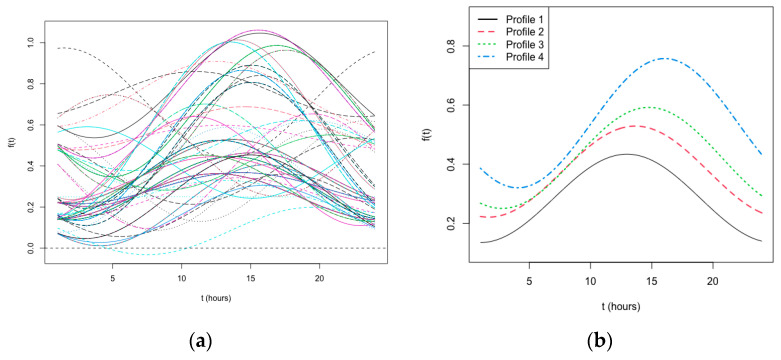
(**a**) Estimated individual chrono-nutritional curves; (**b**) Estimated chrono-nutritional profiles.

**Table 1 nutrients-15-05003-t001:** Distribution of the food contact meals overall (N = 173) by the four estimated chrono-nutritional profiles.

		Chrono-Nutritional Profiles				
Time of Food Contacts	Overall, N = 173	1, N = 80 ^1^	2, N = 11 ^1^	3, N = 55 ^1^	4, N = 27 ^1^	*p*-Value ^2^
**Breakfast**	154 (89%)	74 (92%)	11 (100%)	44 (80%)	25 (93%)	0.10
**Morning Nibbling**	17 (9.8%)	0 (0%)	0 (0%)	8 (15%)	9 (33%)	<0.001
**Morning Snack**	88 (51%)	34 (42%)	11 (100%)	26 (47%)	17 (63%)	0.002
**Lunch**	168 (97%)	80 (100%)	11 (100%)	52 (95%)	25 (93%)	0.067
**Afternoon Nibbling**	38 (22%)	0 (0%)	0 (0%)	22 (40%)	16 (59%)	<0.001
**Afternoon Snack**	108 (62%)	44 (55%)	11 (100%)	32 (58%)	21 (78%)	0.005
**Dinner**	173 (100%)	80 (100%)	11 (100%)	55 (100%)	27 (100%)	-
**Evening Nibbling**	25 (14%)	0 (0%)	0 (0%)	7 (13%)	18 (67%)	<0.001
**Evening Snack**	52 (30%)	0 (0%)	11 (100%)	26 (47%)	15 (56%)	<0.001
**Night Nibbling**	6 (3.5%)	0 (0%)	0 (0%)	6 (11%)	0 (0%)	0.008
**Night Snack**	8 (4.6%)	0 (0%)	0 (0%)	7 (13%)	1 (3.7%)	0.005

^1^ Frequency (%). ^2^ Pearson’s Chi-squared test; Fisher’s exact test.

**Table 2 nutrients-15-05003-t002:** Main characteristics overall (N = 173) according to the three BMI categories based on tertiles.

Characteristics	Overall, N = 173 ^1^	BMI			*p*-Value ^2^
		[32.9, 40.2], N = 58 ^1^	(40.2, 46.2], N = 57 ^1^	(46.2, 80.3], N = 58 ^1^	
**Gender, Female**	117 (68%)	46 (79%)	40 (70%)	31 (53%)	0.011
**Age** (years)	46 (39, 54)	48 (42, 55)	48 (40, 54)	42 (39, 52)	0.11
**Educational level**					0.75
Middle school	72 (42%)	24 (41%)	26 (46%)	22 (38%)	
High school	75 (43%)	24 (41%)	22 (39%)	29 (50%)	
Degree or higher	26 (15%)	10 (17%)	9 (16%)	7 (12%)	
**Physical activity**	29 (17%)	10 (17%)	12 (21%)	7 (12%)	0.43
**Alcohol consumption**	25 (14%)	8 (14%)	8 (14%)	9 (16%)	0.96
**Smoking habit**	33 (19%)	11 (19%)	6 (11%)	16 (28%)	0.067
**Shift work**	32 (18%)	10 (17%)	10 (18%)	12 (21%)	0.87
**Chrono-nutritional profile**					0.45
Profile 1	80 (46%)	32 (55%)	22 (39%)	26 (45%)	
Profile 2	11 (6.4%)	1 (1.7%)	5 (8.8%)	5 (8.6%)	
Profile 3	55 (32%)	16 (28%)	20 (35%)	19 (33%)	
Profile 4	27 (16%)	9 (16%)	10 (18%)	8 (14%)	

^1^ Median (IQR) or Frequency (%). ^2^ Pearson’s Chi-squared test; Kruskal-Wallis rank sum test; Fisher’s exact test.

**Table 3 nutrients-15-05003-t003:** Clinical outcome and psychological scores overall (N = 173) according to the three BMI categories based on tertiles.

	Overall, N = 173 ^1^	BMI			*p*-Value ^2^
		[32.9, 40.2], N = 58 ^1^	(40.2, 46.2], N = 57 ^1^	(46.2, 80.3], N = 58 ^1^	
**Hypertension**	76 (44%)	28 (48%)	25 (44%)	23 (40%)	0.65
**Diabetes**					0.16
No	72 (42%)	26 (45%)	26 (46%)	20 (34%)	
Pre-diabetes	70 (40%)	26 (45%)	22 (39%)	22 (38%)	
Yes	31 (18%)	6 (10%)	9 (16%)	16 (28%)	
**Dyslipidemia**	101 (60%)	28 (48%)	33 (58%)	40 (69%)	0.077
**Use of Statins**	18 (10%)	4 (6.9%)	8 (14%)	6 (10%)	0.46
**CT (mmol/L)**	4.59 (4.06, 5.26)	4.87 (4.36, 5.40)	4.38 (4.01, 5.08)	4.52 (3.93, 4.85)	0.026
**HDL (mmol/L)**	1.17 (1.03, 1.41)	1.33 (1.13, 1.68)	1.14 (0.93, 1.34)	1.11 (0.97, 1.29)	<0.001
**LDL (mmol/L)**	3.08 (2.51, 3.55)	3.11 (2.67, 3.78)	2.98 (2.49, 3.52)	3.07 (2.52, 3.53)	0.48
**TG (mmol/L)**	1.28 (0.93, 1.89)	1.24 (0.86, 1.63)	1.26 (0.91, 1.84)	1.46 (1.14, 2.12)	0.11
**HOMA**	4.0 (2.0, 6.4)	2.5 (1.8, 4.3)	4.1 (2.0, 5.8)	5.9 (3.4, 9.4)	<0.001
**SCL90-R_GSI**	62 (50, 74)	64 (51, 78)	59 (49, 74)	60 (50, 74)	0.27
**SF36PH.**	52 (35, 71)	43 (30, 62)	60 (45, 76)	54 (34, 70)	0.008
**SF36 MH**	55 (39, 72)	52 (31, 68)	56 (41, 73)	56 (44, 69)	0.35
**Y-FAS score**	2 (1, 4)	3 (2, 4)	2 (1, 4)	2 (1, 3)	0.16
**EAT-26 score**	8 (3, 14)	10 (5, 17)	8 (4, 12)	5 (2, 11)	0.017
**BIS-11 score**	59 (53, 67)	64 (57, 67)	58 (53, 66)	58 (52, 68)	0.12
**BES score**	12 (6, 19)	16 (7, 22)	12 (6, 18)	10 (4, 18)	0.11

^1^ Median (IQR) or Frequency (%). ^2^ Pearson’s Chi-squared test; Kruskal-Wallis rank sum test.

**Table 4 nutrients-15-05003-t004:** Odd Ratios (OR) and relative 95% Confidence Interval (95%CI) for the presence of comorbidities estimated by a Logistic regression, with respect to the reference category *.

*Predictors*	Hypertension			Dyslipidemia			Diabetes		
	*OR*	*95%CI*	*p*	*OR*	*95%CI*	*p*	*OR*	*95%CI*	*p*
(Intercept)	0.01	0.00–0.06	**<0.001**	0.77	0.49–1.21	0.265	0.01	0.00–0.10	**<0.001**
Age [+1 years]	1.09	1.06–1.14	**<0.001**				1.05	1.00–1.09	**0.048**
Gender [Male]	2.21	1.08–4.60	**0.031**	11.7	4.90–33.0	**<0.001**	2.99	1.26–7.24	**0.013**
Alcohol cons. [Yes]	0.34	0.12–0.88	**0.032**						
Physical activity [Yes]				2.07	0.83–5.43	0.127	0.31	0.05–1.20	0.139
Smoking habit [Yes]				0.48	0.19–1.16	0.108			
BMI cat. [41.2–46.2]							1.61	0.51–5.39	0.425
BMI cat. [47.2–80.3]							3.12	1.07–10.1	**0.044**

* reference categories: Gender [Female], Alcohol consumption [No], Smoking habit [No], Physical activity [No], BMI category (cat) [32.9–41.1]. The bold refers to the statistically significant values.

**Table 5 nutrients-15-05003-t005:** Adjusted estimated coefficients and relative 95% Confidence Interval (95%CI) of a quantile regression on the physiological outcome, with respect to the reference category *.

*Predictors*	HDL (mmol/L)			TG (mmol/L)			HOMA Index		
	*Est.*	*95%CI*	*p*	*Est.*	*95%CI*	*p*	*Est.*	*95%CI*	*p*
(Intercept)	1.15	0.89–1.41	**<0.001**	0.44	−0.08–0.96	0.097	2.31	1.82–2.80	**<0.001**
Age [+1 years]	0.00	−0.00–0.01	0.088	0.01	0.00–0.03	**0.012**			
Gender [Male]	−0.22	−0.33–−0.11	**<0.001**				1.86	0.59–3.12	**0.005**
Smoking habit [Yes]	0.14	0.02–0.27	**0.026**						
Alcohol consumption [Yes]				0.16	−0.15–0.47	0.299			
BMI cat. [41.2–46.2]	−0.16	−0.32–−0.00	0.051	0.05	−0.24–0.33	0.745	1.51	0.37–2.66	**0.011**
BMI cat. [47.2–80.3]	−0.16	−0.31–0.00	0.053	0.29	−0.03–0.61	0.080	2.82	1.37–4.27	**<0.001**
Chrono-nutr Profile [2]				0.01	−1.03–1.05	0.989			
Chrono-nutr Profile [3]				0.19	−0.12–0.49	0.232			
Chrono-nutr Profile [4]				−0.10	−0.44–0.23	0.551			

* reference categories: Gender [Female], Alcohol consumption [No], Smoking habit [No], BMI category (cat) [32.9–41.1]. Chrono-nutr: chrono-nutritional. The bold refers to the statistically significant values.

**Table 6 nutrients-15-05003-t006:** Adjusted estimated coefficients and the relative 95% Confidence Interval (95%CI) of a quantile regression on psychological scores with respect to the reference category *.

	SF-36 Physical Health					SF-36 Mental Health				SCL-90-R_GSI		
*Predictors*	*Est.*	*95%CI*	*p*			*Est.*	*95%CI*		*p*	*Est.*	*95%CI*	*p*
(Intercept)	69.37	50.98–87.75	**<0.001**			60.07	48.42–71.71		**<0.001**	65.45	57.07–73.82	**<0.001**
Gender [Male]	8.47	−2.61–19.56	0.136									
Age [+1 years]	−0.60	−0.93–−0.28	**<0.001**									
Educational level [High school]	8.51	1.41–15.61	**0.020**									
Educational level [Degree or high.]	4.46	−5.74–14.66	0.393									
Physical Activity [Yes]	6.15	−1.32–13.61	0.109			8.33	−2.67–19.34		0.140			
Alcohol cons. [Yes]	9.59	3.48–15.69	**0.002**			6.60	−7.30–20.50		0.353			
Smoking habit [Yes]						−11.77	−22.52–−1.01		**0.033**	4.30	−2.82–11.43	0.238
BMI cat. [41.2–46.2]	13.35	4.41–22.30	**0.004**			−0.10	−12.67–12.47		0.988	−5.02	−13.83–3.79	0.266
BMI cat. [47.2–80.3]	2.88	−6.04–11.81	0.528			7.63	−3.23–18.50		0.170	−5.38	−13.41–2.66	0.191
Chrono-nutr Profile [2]	−1.27	−9.22–6.69	0.756			0.90	−15.32–17.12		0.914	−6.81	−17.90–4.28	0.230
Chrono-nutr Profile [3]	−9.75	−18.11–−1.40	**0.023**			−12.87	−24.36–−1.37		**0.030**	13.62	5.74–21.50	**0.001**
Chrono-nutr Profile [4]	−12.40	−23.82–−0.98	**0.035**			−3.60	−19.58–12.38		0.659	6.45	−6.14–19.04	0.317
Diabetes [Pre-diabetes]	6.50	−0.94–13.93	0.089							−8.24	−16.28–−0.21	**0.046**
Diabetes [Yes]	8.79	−0.82–18.40	0.075							−7.89	−17.59–1.82	0.113
Dyslipidemia [Yes]	−6.31	−13.15–0.52	0.072			−3.83	−13.72–6.05		0.448			
Observations	173					173				173		
R^2^	0.156					0.008				0.140		
	**EAT-26**			**BIS-11**			**BES**			**Y-FAS**		
** *Predictors* **	** *Est.* **	** *95%CI* **	** *p* **	** *Est.* **	** *95%CI* **	** *p* **	** *Est.* **	** *95%CI* **	** *p* **	** *Est.* **	** *95%CI* **	** *p* **
(Intercept)	7.00	4.81–9.19	**<0.001**	61.00	57.66–64.34	**<0.001**	19.83	12.56–27.10	**<0.001**	2.00	1.39–2.61	**<0.001**
Gender [Male]	−3.33	−5.82–−0.85	**0.009**	3.25	−1.37–7.87	0.170						
Age [+1 years]							−0.13	−0.27–0.02	0.083			
Smoking habit [Yes]	4.67	2.32–7.01	**<0.001**									
Alcohol cons. [Yes]										−1.00	−1.73–−0.27	**0.008**
BMI cat [41.2–46.2]	−1.67	−4.35–1.02	0.225	−3.00	−7.16–1.16	0.159	−4.93	−8.68–−1.17	**0.011**			
BMI cat [47.2–80.3]	−3.33	−5.72–−0.94	**0.007**	−5.50	−9.05–−1.95	**0.003**	−4.01	−7.87–−0.14	**0.044**			
Chrono-nutr Profile [2]	1.00	−4.15–6.15	0.704	1.25	−5.78–8.28	0.728	1.66	−3.73–7.05	0.547	0.00	−1.54–1.54	1.000
Chrono-nutr Profile [3]	2.67	0.33–5.00	**0.027**	4.00	0.53–7.47	**0.025**	5.48	2.10–8.87	**0.002**	1.00	0.15–1.85	**0.023**
Chrono-nutr Profile [4]	4.67	1.44–7.89	**0.005**	4.50	−0.41–9.41	0.074	8.42	4.34–12.50	**<0.001**	1.00	−0.12–2.12	0.083
Hypertension [Yes]	1.67	−0.78–4.11	0.183	3.25	−0.22–6.72	0.068						
Dyslipidemia [Yes]	2.00	−0.50–4.50	0.119				1.94	−1.27–5.16	0.237			
Physical Activity [Yes]				−3.75	−7.09–−0.41	**0.029**	−4.93	−8.44–−1.42	**0.007**			
Diabetes [Pre-diabetes]				−2.75	−5.99–0.49	0.099	−2.93	−6.29–0.43	0.089			
Diabetes [Yes]				−1.00	−6.48–4.48	0.721	−4.09	−8.63–0.45	0.080			
Observations	173			173			173			173		
R^2^	0.123			0.091			0.141			0.007		

* reference categories: Gender [Female], BMI category (cat) [32.9–41.1], Education [Middle school], Alcohol consumption [No], Smoking habit [No], Physical activity [No}, Chrono-nutritional (Chrono-nutr) Profile [1]. Diabetes [No], Dyslipidemia [No] and Hypertension [No]. The bold refers to the statistically significant values.

## Data Availability

Data were extracted from the electronic clinical documentation and patient confidentiality was protected by assigning an anonymous identification code.

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
