# Peer review of "Chrono-Nutritional Patterns, Medical Comorbidities, and Psychological Status in Patients with Severe Obesity"

_nutrients, 2023, doi:10.3390/nu15235003_

Round 1
Reviewer 1 Report
Comments and Suggestions for Authors
Thank you for the opportunity to review this manuscript titled “Chrono-nutritional patterns, medical comorbidities and psychological status in patients with severe obesity” aimed to identify the main temporal eating patterns of patients with obesity, as well as explore the association of these patterns with clinical comorbidities and psychological status.
I have listed a few comments below, which I hope the authors will find useful in revising the manuscript.
· In the method section: how was Physical activity assessed? Also more details regarding the assessment of alcohol consumption and smoking habits are required.
· A particular concern in this study is using the 24-h recall method. Data generated by using 24-h recall might not represent the real picture of the dietary patterns. So, the limitations for using 24-h recall as a dietary assessment method should be acknowledged.
· The differences between snacking and nibbling are not clear as both of them were consumed between meals. The authors need to describe clearly the definition (used in their study) of the three eating occasions (meal, snacking and nibbling).
· Line 146: It has been mentioned that morning nibbling consumed between (9-12 am), do you mean 9 am to 12 pm? Also for lunch and afternoon …etc., Authors should use either 12 Hour Clock or 24 Hour Clock. Do not mix between the two systems to be more clear for the readers.
· Table1: it has been indicated that the p-values were tested by Pearson's Chi-squared test. However, some counts are less than 5.
· The mechanisms supporting findings could be detailed for strengthening the work.
Reviewer 2 Report
Comments and Suggestions for Authors
This study on chrono-nutritional patterns and their potential relation to comorbidities and psychological status in patients with severe obesity is quite interesting. However, there are certain aspects to consider, and limitations that should be noted:
- Sample Specificity: The study focuses on patients with severe obesity. While this provides valuable insights into this specific population, it might not be directly applicable to individuals with different obesity levels or the general population.
- Causation vs. Correlation: The study appears to establish associations (correlations) between chrono-nutritional patterns, comorbidities, and psychological status. It's important to note that correlation does not necessarily imply causation. Further explanation is needed to determine causative relationships.
- Nutritional Patterns: The study identifies four predominant chrono-nutritional patterns. It would be valuable to have more detailed descriptions of these patterns, including the types of foods consumed, portion sizes, and nutrient content. This can help in understanding their nutritional impact.
- Psychological Variables: While the study discusses the influence of chrono-nutritional patterns on psychological variables, it's important to explore the specific psychological conditions or factors under consideration and how they are measured.
In summary, the study offers intriguing insights into the connections between chrono-nutritional patterns, comorbidities, and psychological variables in patients with severe obesity. However, it's important to acknowledge the limitations.
Reviewer 3 Report
Comments and Suggestions for Authors
Review of the manuscript “Chrono-nutritional patterns, medical comorbidities and psychological status in patients with severe obesity”
I have read the manuscript about the evaluation of the chrono-nutritional profiles of 173 patients with severe obesity who were prepared for bariatric surgery. The authors showed the associations of particular chrono-nutritional profiles with age, gender, education, physical activity, smoking habits, BMI classes, glycaemic and lipid profile, and psychological scores of six different questionnaires. Psychological variables were those that seem the most to be positively associated with a constant propensity for eating all day, including night eating. The authors concluded that patients with disorganized, confused, and unbalanced eating patterns and habits are characterized by low perceived quality of life, higher mental distress, the tendency to binge eat presenting symptoms of food addiction, and higher impulsiveness. The study findings confirmed that psychological issues in the development of obesity are of crucial importance due to their relationship with nutritional habits, especially for the evaluation of obesity at the individual level.
Overall, the manuscript has been written well, however, I suggest manuscript editing in the English language. The manuscript could be written in the passive voice, by avoiding using “we” or “our” in the sentences. I have some specific comments and suggestions for improving the manuscript.
1. Write the correct number of patients in the abstract, is it 173?
2. In the Materials and Methods section include information on inclusion/exclusion requirements.
3. Include reference for Friedwald equation (line 89)
4. In the Statistical Analysis section include naming and a reference for statistical software for performing statistical assessments.
5. Sentences in lines 278-280 and in lines 328-331 should be rewritten more clearly and correctly according to the English language style.
6. I suggest rewriting the Discussion section. Perhaps the authors could start by answering the study aims, which are in according to the literature background presented in the Introduction and in the stats of the Discussion. Sentences from lines 354 to 364 could be avoided since most of it was presented in the Results section. It could be discussed here comparing this study's results with those regarding findings from previous similar studies. Also, it could be discussed how this study' detected chrono-nutritional profile could affect or mean to overall health. The mentioned studies in this section could be compared to this study's findings and discussed what they should aim for in future work or investigations or population. Add this study's advantages.
7. In the conclusion the authors could mention future education and consulting of the bariatric patients regarding this study's findings and including their evaluations.
8. All presented table titles should have more information about who as a study group, and the number of patients, and similar because data from tables should be self-explanatory without searching for necessary data in the manuscript.
9. The presented statistical analysis is comprehensive, and the authors put significant effort into all tests, but it is well described.
Comments on the Quality of English Language
I suggest the authors edit the manuscript in the English language.
